



# Multivariate statistical modelling of the drivers of compound flood events in South Florida

Robert Jane[1], Luis Cadavid[2], Jayantha Obeysekera[3], Thomas Wahl[1]

[1]Civil, Environmental and Construction Engineering & National Center for Integrated Coastal Research, University of Central Florida, 12800 Pegasus Drive, Orlando, FL 32816, USA.
[2]Operational Hydraulics Unit – Applied Hydraulics Section, South Florida Water Management District, West Palm Beach, FL 35406, USA.
[3]Sea Level Solutions Center, Florida International University, Miami, FL 33199, USA.

*Correspondence to*: Robert Jane (r.jane@ucf.edu)

**Abstract.** Miami-Dade County (south-east Florida) is among the most vulnerable regions to sea-level rise in the United States, due to a variety of natural and human factors. The co-occurrence of multiple, often statistically dependent flooding drivers – termed compound events – typically exacerbates impacts compared with their isolated occurrence. Ignoring dependencies between the drivers will potentially lead to underestimation of flood risk and under-design of flood defence structures. At present, design assessments of flood defence structures in Miami-Dade County assume rainfall and Ocean-side Water Level (O-sWL) are fully dependent, a conservative assumption inducing large safety factors. Here, an analysis of the dependence between the principal flooding drivers over a range of lags at three locations across the county is carried out. The conservative nature of the existing structural design assessment is subsequently explored, by combining a two-dimensional analysis of rainfall and O-sWL with regional sea-level rise projections. Finally, the vine copula and Heffernan and Tawn (2004) models are shown to outperform five standard higher dimensional copulas in capturing the dependence between the principal drivers of compound flooding: rainfall, O-sWL, and groundwater level. This leads to recommendations for revised future design frameworks able to capture and represent dependencies between different flood drivers.

## 1 Introduction

Florida is more vulnerable to sea-level rise (SLR) in terms of housing and population relative to local mean high tide levels than any other State in the country (Strauss et al., 2012). Miami-Dade County, located in the south-east of Florida, is
particularly vulnerable due to its gently sloped low-lying topography, densely populated coastal areas, and economic importance (Zhang, 2011). Miami, the counties principal metropolitan area, is consistently ranked among the world's most exposed and vulnerable cities to coastal flooding (e.g., Hallegatte et al., 2013; Kulp and Strauss, 2017). While debate surrounds the region's vertical land motion (Parkinson and Donoghue, 2010), the contribution of SLR to nuisance or tidal flooding (Wdowinski et al., 2016) as well as its role in escalating socio-economic impacts such as climate gentrification is becoming
increasingly apparent (Keenan et al., 2018). The future rates of SLR in the region are expected to be greater than the global





average due to variations in the Florida Current and Gulf Stream (Southeast Florida Regional Climate Change Compact, 2015). Higher baseline ocean levels allow storm surges to propagate further inland whilst also reducing pressure gradients in rivers hampering efficient drainage; hence, SLR also increases the fluvial flood potential (Schedel et al., 2018).

In low-lying coastal areas flooding arises because of the interplay between metrological, hydrological, and oceanographic
drivers including rainfall, river discharge, groundwater table, storm surge, and waves. In Maimi Beach, for instance, Wdowinski et al. (2016) found that most flooding events between 1998 and 2013 occurred after heavy rain (> 80 mm) during high tide conditions. The co-occurrence of multiple drivers can exacerbate the impacts of a flood and, depending on the adopted definition, be classified as a compound event (Seneviratne et. al., 2012; Leonard et al., 2014; Zscheischler et al., 2018). For example, significant statistical dependence between heavy rainfall and storm surge (or storm tide) has been identified over a
range of spatial scales: global (Bevaqua et al., 2019), continental (Zheng et al., 2013; Wahl et al., 2015; Paprotny et al., 2018; Wu et al., 2018), regional to national (Svensson and Jones 2002, 2004, 2006; Hendry et al., 2019) and local (Hawkes et al., 2002; Hawkes, 2008; White, 2009; van den Hurk 2013; Lian et al., 2013; Zheng et al., 2014, Bengtsson, 2016). The dependence may arise due to common meteorological forcing (Pugh, 1987), potentially enhanced through orographic effects (Svensson and Jones, 2002, 2004; Martius et al., 2016), or simply by chance (Kew et al., 2013; Martius et al., 2016; Couasnon et al.,
2019). Neglecting even weak dependence can result in the underestimation of water levels (Kew et al., 2013; Zheng et al., 2014; Ikeuchi et al., 2017) and consequently flood risk estimates (Lian et al., 2013; Zheng et al., 2013) in estuarine and tidal channels.

Miami-Dade County is underlined by the highly transmissive and porous (predominantly limestone) Biscayne aquifer which is also the region's main source of potable freshwater (Randazzo and Jones, 1997). The lateral intrusion of saltwater into the
unconfined aquifer as a recirculating "saltwater wedge" is widely acknowledged (Provost et al., 2018). SLR along with an increased likelihood of recurring drought during the winter-spring season, associated with changes in the climate system, enhances the risk of contamination of the water supply (Bloetscher et al., 2011). Furthermore, the County's population is expected to increase by nearly 20% in the next 20 years (Bureau of Economic and Business Research, 2015), increasing flood exposure and demand on water resources. The South Florida Water Management District (SFWMD) is responsible for
managing and protecting the water resources of South Florida. The SFWMD must balance demand for potable water and agricultural and landscape irrigation with flood mitigation, whilst ensuring the water table remains sufficiently high to prevent saltwater intrusion and achieve other ecological objectives (SFWMD, 2016). Their aim is to meet these objectives through the continuous operation of an extensive network of drainage canals, storage areas, pumps, and other control structures. The Biscayne aquifer has a direct hydraulic connection to the natural and man-made surface water bodies, a consequence of its
shallow depth and high porosity, and is therefore considered a part of this integrated hydrologic system (Randazzo and Jones, 1997).

In heavily managed urbanized catchments, antecedent groundwater conditions are an essential initial condition for hydraulic/hydrological models for robust flood risk analysis (Hettiarachchi et al., 2019). Rainfall is often employed as a surrogate for river discharge (e.g., Zheng et al., 2013; Wahl et al., 2015; Bevacqua et al., 2020). Physical properties such as





the size, gradient, and permeability of a catchment influence the river response to a given rainfall event (Svensson and Jones,
      2002; Zheng et al 2013; Hendry et al., 2019). Verhoest et al. (2010) demonstrated that the return period of a rainfall event may
      differ significantly from that of the corresponding discharge, depending on the antecedent wetness of a catchment. In south-
      east Florida, approximately half of the average annual rainfall is lost to evapotranspiration (Bloetscher et al., 2011), hence
      rainfall is unlikely to constitute a suitable proxy for discharge.

Due to the unusually high connectivity of ground and surface water hydrology, south-east Florida has a high propensity for
      pluvial flooding. The concurrence of heavy precipitation and high antecedent soil moisture is the dominant flood generating
      mechanism for most catchments without significant snow melt (Berghuijs et al., 2016, 2019). Many recent studies (Moftakahri
      et al., 2017, 2019; Bevaqua et al., 2017; Couasnon et al., 2018, 2019; Paprotny et al., 2018; Ward et al., 2018; Serafin et al.,
      2019) statistically model river discharge and surge (or coastal water level in the case of Ganguli and Merz (2019)), or their

relevant proxies (Kew et al., 2013), as opposed to rainfall and surge, implicitly accounting for catchment properties and pre-
      existing groundwater level (Lamb et al., 2010). Not accounting for groundwater level explicitly, especially in areas like Miami-
      Dade County where groundwater levels are highly responsive (and potentially correlated) to rainfall and O-sWL, precludes a
      robust assessment of the risk of pluvial flooding. Therefore, in this work, statistical models will be tested for their ability to
      capture the joint probability distribution of rainfall, O-sWL (tide + non-tidal residual), and groundwater level.

Traditional multivariate probability distributions are often restrictive in terms of the choice of marginal distributions, i.e., all
      the margins are required to be the same type of distribution. For example, fitting a bivariate Gaussian distribution to extreme
      tides and corresponding freshwater flows required Loganathan et al. (1987) to assume Gaussian marginal distributions.
      Copulas allow the dependence and marginal modelling to be carried out independently providing more flexibility in the choice
      of marginal distributions than traditional multivariate models (Patton, 2009). Consequently, bivariate copulas have been used

extensively in the modelling of compound flooding induced by rainfall and surge (e.g., Wahl et al., 2015) and from discharge
      in multiple rivers at their confluences (Wang et al., 2009; NCHRP 2010; Chen et al., 2012; Bender et al., 2016; Peng et al.,
      2017, 2018, Gilja et al., 2018). Higher-dimensional multivariate parametric copulas are limited in the sense that they assume
      homogeneity in the type of dependence between each pair of variables (Aas et al., 2009). Pair Copula Constructions (PCC's)
      in contrast take advantage of the rich array of bi-variate copulas and overcome this limitation by decomposing higher

dimensional probability density functions (pdf's) into a cascade of bi-variate copulas (Bedford and Cooke, 2002). Bevacqua
      et al. (2017) implemented PCC to model the conditional joint pdf of river discharge and sea levels (given meteorological
      predictors) to assess compound flood risk in Ravenna, Italy. The method proposed by Heffernan and Tawn (2004; referred to
      hereafter as HT04) is an alternative to higher-dimensional multivariate parametric copulas requiring no assumptions regarding
      the type of dependence between variable pairs.

In current structural design assessments, the SFWMD assumes full dependence between rainfall and O-sWL. Consequently,
      any correlations <1 between the drivers will potentially lead to an overestimation of risk and conservative design. The overall
      aim of the paper is to assess the different drivers of compound flooding in coastal areas of Miami-Dade County. This will be
      achieved by meeting three objectives. The first objective is to determine whether there is any statistically significant correlation



between extreme rainfall, O-sWL, and groundwater level, while accounting for relevant time-lags. The second objective is to
assess the conservative nature of the present structural design approach. This includes a bivariate statistical analysis, akin to
those in previous studies, but also including regional SLR scenarios to assess how long it will take for any safety-margin (that
is implicitly included by assuming full dependence between drivers) to be exhausted. The third and final objective is to
incorporate antecedent catchment conditions into the statistical model and to provide robust estimates of the joint probabilities
(using a variety of approaches) of extreme rainfall, O-sWL, and groundwater table.

## 2 Case study sites and data

Miami-Dade is situated in south-east Florida (Figure 1a). The Everglades Water Conservations Areas comprise the western
portion of the County whilst heavy engineered water infrastructure and flood control systems have facilitated agricultural and
urban development farther east. Three case study sites, differentiated by the colors in Figure 1b (named after the structures
where O-sWL is measured), were selected to allow an assessment of the variation of the hydrological behavior with latitude.
The study is undertaken using in-situ observations with each site defined by a rainfall gauge, stage gauge (to measure the O-
sWL), and groundwater well.

Rainfall data consists of daily precipitation totals obtained from the National Oceanic and Atmospheric Administration's
(NOAA's) National Climatic Data Center's archive of global historical weather and climate data. The rainfall record at Miami
Airport is complete, while the records at Perrine and Miami Beach contain a substantial number of missing values; constituting
22.85% and 4.80% of the total time series, respectively. The highly localized nature of individual rainfall events in the region
along with the spatial and temporal resolution of rainfall measurements renders the estimation of missing daily rainfall values
using neighbouring gauges impractical (Pathak, 2001).

Stage gauges are attached to flood/salinity control structures, operated by SFWMD to maintain the water level to prevent
saltwater intrusion and release canal water to the sea (typically via tidally modulated channels) alleviating potential flooding.
The stage time series downstream of the relevant structures (here termed O-sWL) were extracted from DBHYDRO (SFWMD's
corporate environmental database) and converted to daily maxima. O-sWL refers to the still water level (i.e., the water level
discounting waves/wave set-up) that comprises mean sea level, the astronomical tidal component, and non-tidal residual (Pugh,
1987).

Groundwater wells (maintained by the United States Geological Survey) closest to each stage gauge and with record lengths
similar to the O-sWL time series were identified and daily maximum water level records extracted from DBHYDRO. An
analysis of the distribution of the missing O-SWL and groundwater observations indicated the presence of long gaps in some
of the records, prohibiting linear interpolation of the record to infill missing values. However, both the O-sWL and groundwater
records showed a high degree of linear correlation with corresponding records at nearby sites. Missing values were therefore
imputed through a linear regression of the observations at the location of interest on those at nearby sites (Figure SM.1),
starting with the closest site to the location of interest. Any remaining non-consecutive missing values were imputed through
linear interpolation (See Figures SM.2-SM.6).

**Figure 1. Study site location and data completeness. (a) Miami-Dade County in the state of Florida, USA. (b) Topographical map of the eastern portion of the County showing the location of the measuring stations for the three**
**case study sites. Principal stations are named whilst those used for data imputation are not labelled. (c) Completeness of the records at each site's three principal stations along with the method adopted to impute specific missing values.**

A fundamental assumption of the standard extreme-value theory statistical models is that the analysed data sets consist of Independent and Identically Distributed (IID) random variables. The models thus require stationarity, i.e., the statistical parameters such as mean and variance should remain constant over time and be free of "trends, shifts, or periodicity" (Salas,
1993). It is standard practice to transform the data to achieve stationarity through detrending (e.g., Wyncoll et al., 2016). The long-term mean sea level signal is superimposed onto inter-annual to multi-decadal sea level variability caused by tidal modulations associated with the nodal (18.61 year) and perigean (8.5 year) cycles, and other oceanic-atmospheric processes (e.g. Valle-Levinson et al., 2017). Here, a moving window approach is applied to the O-sWL series to remove long-term sea level rise and seasonality effects (Arns et al., 2013). In the procedure, the estimate of the trend is subtracted from the original
time series value yielding a residual, which is then added to the mean sea level derived from the last five years of data to





represent most recent mean sea level conditions. The groundwater level was detrended in an identical manner. The detrended series are shown alongside the imputed observational records in Figures SM.7 to SM.12.

## 3 Methodology

Section 3.1 introduces the measures for assessing the strength of the dependence between the drivers and identifying the type

of dependence in their joint tail regions. Section 3.2 describes methods employed for the bivariate analysis of rainfall and O-sWL, before the choice of hazard scenario is scrutinized. Finally, Section 3.3 provides a description and justification for the statistical models adopted for the trivariate analysis including groundwater level.

### 3.1 Dependence analysis

Kendall's rank correlation coefficient $\tau$ provides a measure of the degree of the association between the variables. As opposed

to linear correlation, rank correlation is able to capture any non-linear relationships between a pair of variables, whist $\tau$ possesses several desirable properties over other rank correlation measures (Li et al., 2012). The value for each pair of variables will also be used here to determine whether there is statistically significant correlation between them, i.e., if the null hypothesis $H_0: \tau = 0$ can be rejected.

Extremal dependence falls into one of two classes: asymptotic dependence or asymptotic independence (Ledford and Tawn,

1997). If $(X, Y)$ are a pair of variables with distribution functions $(Fx, Fy)$ transformed to common uniform $(0,1)$ distributions, i.e. $(U = Fx(X), V = Fy(Y))$, an intuitive measure of the extremal dependence of $(X, Y)$ is $\chi$ (Buishand 1984 and Coles et al., 1999):

$$\chi = \lim_{u \to 1} P(V > u | U > u) , \tag{1}$$

where $P(A|B)$ is the conditional probability of A given B. For independent variables $\chi = 0$, for asymptotically dependent

variables $\chi$ increases with dependence strength, and $\chi = 1$ signals perfect dependence. To obtain $\chi$ it is more convenient to consider

$$\chi(u) = 2 - \frac{\ln[P(U > u, V > u)]}{\ln[P(U > u)]} \tag{2}$$

an asymptotically equivalent function, i.e. $\chi = \lim_{u \to 1} \chi(u)$, for $0 \leq u \leq 1$. Coles et al. (1999) introduced a second measure $\bar{\chi}$ to quantify the magnitude of the dependence between a pair of asymptotically independent variables:

$$\bar{\chi}(u) = \frac{2 \ln[P(U > u)]}{\ln[P(U > u, V > u)]} - 1 \tag{3}$$





where $-1 < \bar{\chi} \leq 1$, for $0 \leq u \leq 1$ and $\bar{\chi} = \lim\limits_{u \to 1} \bar{\chi}(u)$. In the case of full dependence $\bar{\chi} = 1$, whilst for the class of asymptotically independent variables $\bar{\chi}$ increases with dependence strength. Empirical estimates of $\chi(u)$ and $\bar{\chi}(u)$ are possible by approximating the probabilities in equations 2 and 3 with the equivalent proportions observed in the data.

Svensson and Jones (2002) proposed a bootstrap procedure to test for asymptotic dependence. The two records are independently sampled with replacement using a sample size the same length as the original concurrent record. The samples are subsequently paired to create a dataset identical in size to the original but with the dependence removed. The process is repeated to create a large number $N$ of datasets. For each data set $\chi$ is calculated and denoted by $\chi_{Boot_i}$, $i = 1, \dots, N$. If less than 5% of $\chi_{Boot_i}$ are greater than the estimate of $\chi$ associated with the observed data, then there is strong evidence against the null hypothesis $H_0 \colon \chi = 0$. Zheng et al. (2013) used the procedures described here to assess the asymptotic behavior of rainfall and storm surge along the Australian coastline, detecting the presence of both dependence classes.

### 3.2 Bivariate analysis

Here, a two-sided sampling approach similar to that in Wahl et al. (2015) is implemented to identify bivariate extreme events. Due to the relatively short length of the overlapping records and wastefulness of the block maxima approach the threshold exceedance method is first used to identify univariate extremes. In practice, the method of Smith and Weissman (1994) is applied to the rainfall time series to identify cluster maxima which are paired with simultaneous O-sWL values and vice versa to create two 2-dimensional time series.

A copula is a multivariate probability distribution with uniform marginal distributions. If $X_1, \dots, X_d$ are a set of $d$ continuous random variables with joint distribution function $F_{X_1, \dots, X_d}(x_1, \dots, x_d)$ then according to Sklar's theorem (Sklar, 1957) there exists a unique copula $\boldsymbol{C}$ on $[0,1]^d$ such that

$$\boldsymbol{F}_{X_1, \dots, X_d}(x_1, \dots, x_d) = \boldsymbol{C}\left(F_{X_1}(x_1), \dots, F_{X_d}(x_d)\right) \tag{4}$$

where $F_{X_i}$ is the marginal distribution of $X_i$, $i = 1, \dots d$. Hence, any multivariate joint distribution can be decomposed into the set of univariate marginal distributions and a copula. The latter contains all the information about the dependence structure of the joint distribution.

For a range of thresholds, the best-fitting of 40 competing copulas plus the independence copula are determined via the Akaike Information Criterion (AIC), using the *VineCopula* R package (Schepsmeier et al., 2018). For the conditioned variable, cluster maxima above a sufficiently high threshold are fitted to a Generalized Pareto Distribution (GPD). The marginal non-conditioned variables are modelled by parametric distributions. Two unbounded continuous distributions are fitted to O-sWL in the sample where rainfall is conditioned to exceed a predetermined threshold. A range of continuous distributions supported on $[0, \infty)$ are fitted to rainfall in the sample where O-sWL is conditioned to exceed a predetermined threshold. In each case, several parametric tests and diagnostic plots are subsequently utilized to determine the best fitting marginal distribution (see Supplementary Material for more details).



As opposed to the univariate case where the region containing "dangerous" events is uniquely defined, in the bivariate and higher dimensional settings hazard scenarios are required to specify this region. For a $d$-dimensional probability distribution function $\boldsymbol{F} = \boldsymbol{C}(F_1, \ldots, F_d)$ and $\alpha = (0,1)$, Salvadori et al. (2011) define the critical layer $L_\alpha^F$ of level $\alpha$ as the following set

$$L_\alpha^F = \{\boldsymbol{x} \in R^d : \boldsymbol{F}(\boldsymbol{x}) = \alpha\}. \tag{5}$$

The critical layer is an iso-hyper-surface of dimension $d - 1$. Thus, it corresponds to an (iso)line (also referred to as a contour line) in the bivariate case and to a (iso)surface in the trivariate case. Each critical layer partitions $R^d$ into three non-overlapping exhaustive regions: a super critical layer comprising the events considered "dangerous", the critical layer itself, and a subcritical layer containing "safe" events.

There are several definitions of hazard scenarios, including "OR", "AND", Kendall (Salvardori et al., 2004), and survival Kendall (Salvadori et al., 2013), each offering different perceived strengths and limitations (e.g., bounded vs. unbounded subcritical layer, mathematical vs. physical valid interpretation) (e.g., Salvadori et al., 2011; Gräler et al., 2016). Due to the absence of any physical interpretation, Salavadori et al. (2016) suggest the procedures à la Kendall be reserved for preliminary assessments to gauge the expected probabilities of multivariate occurrences. The "OR" scenario has been extensively applied

in the context of compound flooding at river confluences (e.g., Wang et al., 2009; Bender et al., 2016). Recently, Moftakhari et al. (2019) proposed incorporating the "AND" scenario to estimate the joint return period of river discharge and ocean levels in the FEMA (2015) procedure for assessing compound flood hazard in tidal channels and estuaries. In line with this recommendation (and many other previous applications where ocean levels and pluvial/fluvial flood drivers were analysed) the "AND" hazard scenario is adopted in this study.

A methodology for deriving design events when adopting a conditional sampling method with two joint probability distribution functions, as proposed in this paper, is put forward and implemented in Bender et al. (2016). The approach exploits the strict monotonicity of the joint distribution functions, by defining the (quantile-) isoline functions, for level $\alpha$, implicitly as $F_{O-sWL|R}(x_R, q_{O-sWL|R}(x_R)) = \alpha$ and $F_{R|O-sWL}\left(x_R, q_{R|O-sWL}(x_R)\right) = \alpha$. The possible design events comprise the outer envelope created by overlapping the two isolines, i.e. $x \mapsto \max\{q_{O-sWL|R}(x_R), q_{R|O-sWL}(x_R)\}$ (see Figure 2 for a hypothetical

example illustrating the approach).




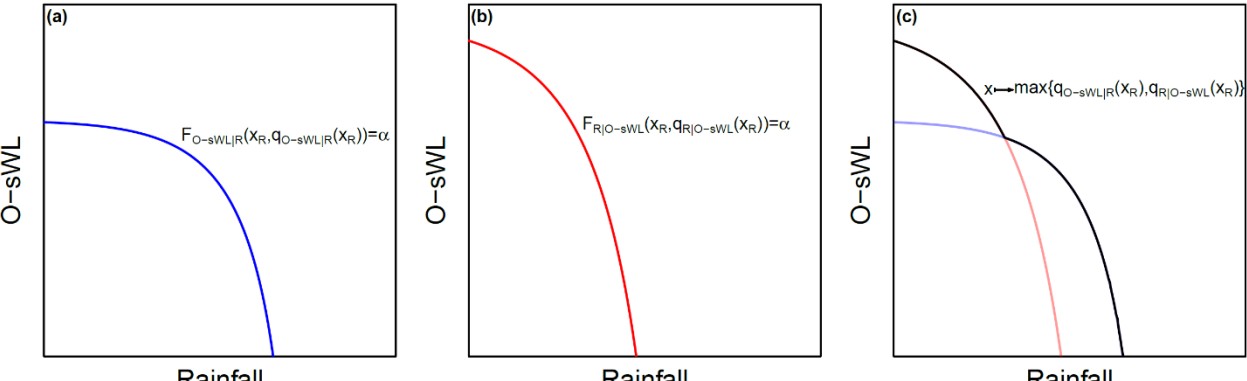

**Figure 2. Schematic illustrating the approach by Bender et al. (2016) for combining two isolines of level** $\alpha$. **The (quantile-) isolines from the joint distributions conditioning on (a) Rainfall and (b) O-sWL, respectively. (c) A single (quantile-) isoline is given by the envelop created by overlapping the isolines in (a) and (b).**

The choice of hazard scenario should reflect the type of dangerous event, e.g., a mechanism of failure, but is often an arbitrary and subjective choice (Serenaldi 2015, Gouldby et al., 2017). Volpi and Fiori (2014) noted the typical disparity in the return period of structural failure compared with that of the loading variables and consequently proposed the so-called structure-based return period. The structure-based return period is derived by propagating the joint distribution of the basic variables through a structure or response function, describing the physical dynamics of a system. Hence, the return period of a response

variable is calculated directly, typically empirically from a (large) sample of the basic variables after fitting a multivariate statistical model (Gouldby et al., 2017). The approach thus negates the need for a practitioner to define a hazard scenario (Salvadori et al., 2016). Serenaldi (2015) argues the concept of return period in univariate frequency analysis is prone to misconceptions, only exacerbated in the multidimensional domain, and that the risk of failure offers a more transparent and suitable measure of risk. A full risk analysis is beyond the scope of this study, but recommended as future work.

**3.3 Trivariate analysis**

This section provides a description of three types of multivariate statistical models – standard higher dimensional copulas (Section 3.3.1), Pair Copula Construction (Section 3.3.2) and the HT04 model (Section 3.3.3) – applied here to capture the dependence between extreme rainfall, O-sWL, and groundwater levels.

**3.3.1 Standard higher dimensional copulas**

Copulas were first introduced to the field of hydrology in De Michele and Salvadori (2003), where an Archimedean copula was used to describe the dependence between storm duration and average rainfall intensity. The Archimedean copula family comprises a rich array of radially asymmetric and symmetric copulas covering a diverse range of upper and lower tail dependence. The strengths of all pairwise dependencies are captured by a single parameter, thus standard Archimedean copulas are symmetric for any permutation of indexes (exchangeability). The exchangeability of Archimedean copulas is often



considered strongly restrictive in higher dimensional applications, as it implies all pairwise dependencies are identical (Di
Bernardino and Rullière, 2016). Elliptical copulas, as the name suggests, are simply copulas of elliptical distributions and
consequently possess many of the useful traceable properties of these multivariate distributions (Fang et al., 1990). Elliptical
copulas are radially symmetric with a correlation matrix of parameters describing the strength of the pairwise dependencies.
Consequently, they are non-exchangeable, only assuming the type of dependence within each tail are identical. The trivariate

Gaussian copula $C^{Gauss}$ is given by,

$$C^{Gauss}(u_1, u_2, u_3) = \boldsymbol{\Phi}_R \left( \Phi^{-1}(u_1), \Phi^{-1}(u_2), \Phi^{-1}(u_3) \right) \tag{6}$$

where $\boldsymbol{\Phi}_R$ is the joint cumulative distribution function (CDF) of the standard trivariate normal distribution with correlation
matrix $R$, and $\Phi^{-1}$ is the inverse CDF of the univariate standard normal distribution. The Student's $t$ copula also possesses a
degrees of freedom parameter $v$ specifying the additional probability density assigned to the joint tails compared with the

Gaussian copula *ceteris paribus*. The Student's $t$ copula approaches the Gaussian copula as $v \rightarrow \infty$. In contrast with the
Student's $t$ copula, which possesses tail dependence, the Gaussian copula assumes asymptotic independence, i.e. the Gaussian
copula has zero tail dependence $\chi = 0$.

Whilst bi-variate applications are extensive in hydrology, trivariate applications of standard copulas are scarce. From analysing
the dependence between drought duration, intensity, and severity in New South Wales (Australia), Wong et al. (2008) found

the Gumbel copula outperformed the Gaussian copula. In a similar application in the Weihe River basin (China), Ma et al.
(2010) reported that the trivariate Gaussian copula gave a better fit than the Student's $t$ copula, with both outperforming six
Archimedean copulas, half of which possessed (radial) asymmetry. In other environmental applications, the Student's $t$ copula
has been shown to offer a superior fit to the Gaussian copula in the presence of tail dependence (e.g., Jane et al., 2016; Wahl
et al., 2016). In this study two elliptical copulas and three Archimedean copulas (Gumbel, Clayton, and Frank) are considered.

The three Archimedean copulas comprise a range of tail dependence regimes, i.e., upper, lower, and no tail dependence, and
are consequently commonly applied together to assess the type of dependence between a set of variables (e.g., Daneshkhah et
al., 2016).

### 3.3.2 Pair-Copula Constructions

Approaches to increase the flexibility of standard higher dimensional copulas include techniques to remove the exchangeability

property of Archimedean copulas (e.g., Di Bernardino and Rullière, 2016) as well as the development of (meta-elliptical)
copulas for various meta-elliptical distributions (Fang et al., 2002). Pair-copula construction (PCC) provides greater flexibility
and a more intuitive way of extending bi-variate copulas to higher dimensions than these approaches (Aas et al., 2009).

PCC, originally proposed by Joe (1996), decomposes a $d$-dimensional probability distribution into the product of a cascade of
bivariate copulas and the marginal densities of each variable. PCC permits the free specification of $\frac{d(d-1)}{2}$ copulas; the first

$d - 1$ copula densities are dependence structures of unconditional bivariate distributions while the remaining are of conditional

bivariate distributions. As $d$ increases, the number of mathematically equally valid decompositions soon becomes large. To ensure consistent definitions of each distribution in a PCC, Bedford and Cooke (2001, 2002) introduced the regular vine, a graphical model for specifying the conditional dependencies in a decomposition. In the $d-$dimensional case a vine consists of a set of $d-1$ nested trees. The edges of tree $T_j$ become the nodes of tree $T_{j+1}$, $i = 1, \dots, d-2$, where nodes represent the

variables, and the labels of each edge denote the subscript of a pair-copula. A regular vine is a vine in which two edges are joined in tree $T_{i+1}$ only if they share a common node in tree $T_j$.

The class of regular vines is considered relatively broad and encompasses a range of possible pair-copula decompositions. The canonical (or C-) vine and D-vine are special cases of regular vines, defining specific ways of decomposing a multivariate probability density. Each of the three possible decompositions of a three-dimensional copula density are simultaneously both

a C- and a D-vine (see Figure 3 for one example).

Gräler et al. (2013) applied a bivariate copula to annual maximum peak discharge and its volume, as well as a trivariate vine copula, by also including duration, to investigate the effect of different modelling choices on design events. They found evidence of design quantiles shrinking as the number of variables considered grows (bivariate vs. trivariate) referred to as the *dimensionality paradox* (Salvadori and De Michele, 2013). They concluded that practitioners should strive for a balance

between the number of variables considered and (numerical) complexity of the copula. In a similar study, Daneshkhah et al. (2016) showed a vine copula outperformed five tested standard higher dimensional multivariate copulas.

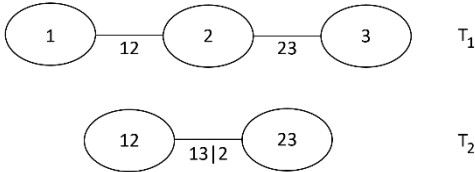

**Figure 3: General structure of three-dimensional C/D- vine copula**

An alternative pair-copula decomposition of a higher dimensional joint probability distribution is the Nested Archimedean Construction (NAC) (e.g., Embrechts et al., 2003). In NAC, only $d-1$ copulas are user specified, whilst the remaining copula and parameters are defined implicitly through the construction. In addition, the bi-variate copulas are required to be Archimedean copulas and there are strong restrictions on the parameters. After the application of PCC and NAC to two four-dimensional financial and environmental datasets, Aas and Berg (2009) concluded that PCC is superior both in terms of

goodness-of fit as well as computational efficiency.

### 3.3.3 Heffernan and Tawn (HT04) approach

The HT04 approach models the conditional distribution of the remaining variables given a specified variable exceeds a suitably high threshold. By repeating the procedure for each variable in turn the model captures the dependence structure between a set





of variables when at least one takes on an extreme value. The HT04 approach thus requires no assumptions regarding the nature of the dependence in the joint tail regions between a set of variables.

As opposed to the standard copula methodology, the HT04 model is generally implemented using Gumbel marginal distributions given by $Y_i = -\log(-\log[\hat{F}_i(X_i)])$, where $\hat{F}_i$ is an estimate of the cumulative distribution function of $X_i$. Alternative scales can be invoked to transform the data to common marginals. For instance, Keef et al. (2013) describe the advantages of using Laplace scales, particularly if any variables exhibit a negative association. To remain consistent with HT04 and most other applications of the approach, Gumbel scales are adopted in this work. If $\boldsymbol{Y}_{-i}$ is the vector of all the (transformed) variables excluding $Y_i$ the HT04 model is typically implemented via the following multivariate non-linear regression model

$$\boldsymbol{Y}_{-i}|Y_i = \boldsymbol{a}Y_i + Y_i^{\boldsymbol{b}}\boldsymbol{Z} \text{ for } Y_i > \upsilon \tag{7}$$

where $\upsilon$ is a suitably high threshold on $Y_i$, $\boldsymbol{a}$ and $\boldsymbol{b}$ are vectors of parameters and $\boldsymbol{Z}$ is a vector of residuals. The parameters $\boldsymbol{a}$ and $\boldsymbol{b}$ are estimated using maximum likelihood under the temporary assumption that $\boldsymbol{Z}$ is normally distributed with unknown mean and variance. Recently, Tove et al. (2019) removed the temporary Gaussian assumption on the joint residual distribution, by instead modelling the distribution semi-parametrically using a Gaussian copula and kernel density estimated marginals. This alteration permits new combinations of $\boldsymbol{Z}$ to arise thus enabling non-deterministic extrapolation of past events; in the context of the present study this is to be considered in future work.

An outline of the steps involved in the well-established Monte-Carlo procedure for generating a realization $\boldsymbol{Y}$ (on the transformed scale) from the fitted model is given below (e.g,. Keef et al., 2009a; Gouldby et al., 2014):

1. Sample $Y_i$, conditional on $Y_i > \upsilon$.
2. Independently sample a joint residual $\boldsymbol{Z}$.
3. Calculate $\boldsymbol{Y}_{-i}$, from equation (9) using relevant regression parameters, $Y_i$, $\boldsymbol{Z}$.
4. Reject sample $\boldsymbol{Y}$, unless $Y_i$ is a maximum

Given the desired sample dimension, the sequence of steps is repeated until the expected number of events where variable $Y_i$ is a maximum, conditioned to exceed the threshold, is consistent with the empirical distribution. The procedure is repeated, conditioning on each variable in turn to ensure the appropriate proportion of events are simulated. The sample can then be transformed to original scales using the marginal distributions and the inverse probability integral transform.

The extremes observed during such temporal dependent and spatially varying events may not occur concurrently. Keef et al. (2009a) addressed this limitation by fitting the HT04 model to the distribution of the variable at location $j$ at a lag of $\tau$ in relation to an extreme value observed at location $i$, i.e., the model is fitted to $Y_{j,t+\tau}|Y_{i,t}$ for $Y_{i,t} > u$, for a range of $\tau$ and each $i \neq j$. Subsequently, Keef et al. (2009b) applied the method to investigate the spatial dependence of rainfall and river flow in Great Britain and found that both types of extreme events become increasingly localized with increasing return period. Similarly, multi-site, single variable applications are common in the literature (e.g., Lamb et al., 2010; Diederen et al., 2019). The model has also been applied to capture the dependence in the variables contributing to extreme sea states at a single location (e.g., Gouldby et al., 2014) and at multiple sites (Wyncoll et al., 2016).





## 4. Results

In this section, the results of the correlation analysis (Section 4.1), bivariate analysis (Section 4.2), and trivariate analysis (Section 4.3) are discussed in turn. In Sections 4.2 and 4.3 results pertain to Site S22, analogous results for the other two sites
are provided in the Supplementary Material.

### 4.1 Dependence analysis

Rainfall, O-sWL, and groundwater level exhibit small ($\tau < 0.3$) but generally statistically significant correlations over a range of time-lags (Figure 4). The strength of the correlation of rainfall and O-sWL with groundwater level decreases with distance north across study sites. In addition, the peaks in the correlations are situated at zero lag at site S20, but at lags in the
groundwater level of between two and four days at sites S22 and S28, respectively. The peaks in these correlations also become increasingly shallow with distance north, indicating that the water table is more responsive at S20 than at the other sites. This is likely a consequence of the lower elevation at S20, resulting in the water table typically laying closest to the ground surface. Rainfall and O-sWL are the least correlated of the variable pairs, exhibiting little variation in correlation strength or variation with lag between the sites.
The empirical estimates of $\chi(u)$ and $\bar{\chi}(u)$ as $u \rightarrow 1$ (Figure 4, middle and bottom) provide an informal assessment of the asymptotic behavior of the joint distribution of the drivers. The informal analysis failed to provide conclusive evidence of asymptotic dependence or asymptotic independence, an issue also highlighted in Coles et al. (1999). For instance, at site S20 the best estimate of $\chi$ for each pair of drivers is positive once $u > 0.4$ indicating asymptotic dependence. However, the confidence intervals for $\chi$ always include zero and $0 < \bar{\chi}(u) < 1$, contradicting the conclusion of asymptotic dependence. On
the other hand, all pairs of drivers at all sites returned statistically significant results in the hypothesis test proposed in Svensson and Jones (2002), providing strong evidence against the null hypothesis of asymptotic independence.


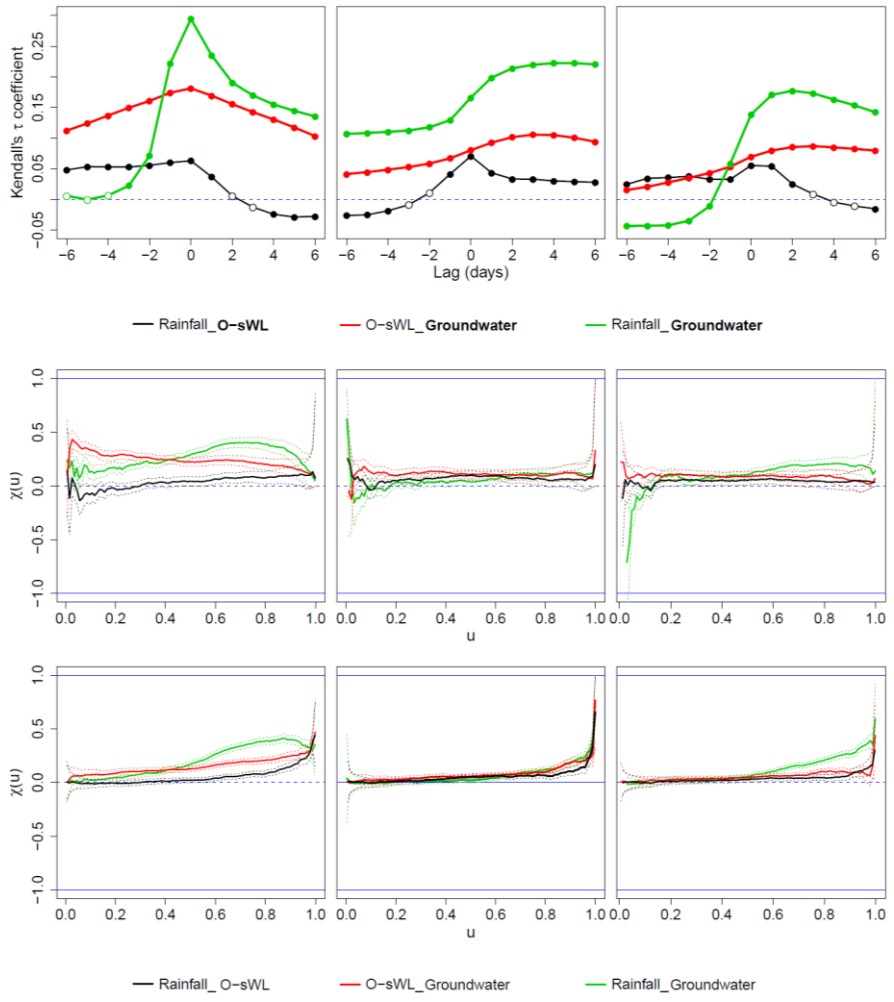

**Figure 4. Assessment of correlation between the flooding drivers at Site S20 (1st column), S22 (2nd column) and S28 (3rd column). Top: measure of the pairwise association $\tau$ between the drivers over various lags. Filled dots indicate the presence of statistically significant correlations (p-value<0.05). The lag is applied to the quantity shown in bold. Middle and Bottom: estimates of $\chi(u)$ and $\overline{\chi}(u)$ along with the associated 95% confidence intervals.**

### 4.2 Bivariate analysis

To capture the dependence between rainfall and O-sWL, the approach outlined in section 3.2 was applied for a range of thresholds. The choice of copula family is relatively insensitive to the selected threshold (see Figures SM.13 to SM.15). The threshold is selected as a trade-off between the bias and variance in the copula parameter estimates. For each of the conditioned samples a threshold of the 0.98 quantile of the conditioning variable was deemed appropriate at each of the sites. Attention from hereon in focuses on site S22 (detailed results for the other sites are included in the Supplementary Material) where the



0.98 quantile threshold gives an average of 6.3 and 5.2 events per year when conditioning on O-sWL and rainfall, respectively.
The conditioning variable was fitted to a GPD while relevant non-extreme parametric distributions were fitted to the non-conditioning variable. The Birnbaum-Saunders(logistic) distribution was selected to model the rainfall(O-sWL) data in the sample where O-sWL(rainfall) is conditioned to exceed its 0.98 quantile, as it was consistently among the best fitting of the candidate distributions at the three sites (see Figures SM.16 to SM.18).

The quantile isolines for several return periods are shown alongside the observations in Figure 5. The coloured contours on the isolines represent the relative likelihood of events. The "most likely" strategy is used as a simple way to derive possible design events associated with a given return period $T$ (Salvadori et al., 2011, 2013). Practically, the design event is given by the point of maximum relative probability density on the isoline associated with return period $T$. In this work, the relative probabilities are estimated non-parametrically via a Kernel Density Estimate (KDE), using the *ks* R package (Duong, 2007). Initially KDE was applied to the observations, however, particularly for larger return periods the design event proved highly sensitive to a small number of observations. Hence, design events were determined by applying KDE to a large sample $N = 10,000$ from the two fitted copulas, with sample proportions consistent with the empirical distributions, and transformed back to original scales.

The upper and lower panels of Figure 5 illustrate two types of design events, indicating that the system experiences a change in behaviour between 20- and 50-year return periods. To further investigate the return period at which the change in design event type occurs, design events were calculated for return periods from 1 to 100 years at a yearly interval. The processes of simulating samples from the fitted copulas, estimating the relative likelihood along the isolines and extracting the "most-likely" event was then repeated to give 100 design events associated with the 1- to 100-year return periods. The results showed that the change occurs for return periods between approximately 20 and 40 years. For small return periods ($\leq$20 years), design event rainfall remained <1mm, thus they may be considered "surge only" events. Consequently, the existing SFWMD design events are only marginally conservative in terms of O-sWL, yet highly conservative with respect to rainfall. For return periods, greater than say 40 years, design events resemble compound events. As the return period increases, rainfall values given by the bivariate approach increasingly resemble the corresponding univariate return period rainfall (i.e., the most-likely event moves to the right along the x-axis). Conversely, the O-sWL of the design event given by the new and existing approaches diverge as return periods increase (i.e., the most likely design-event moves down along the y-axis). For instance, the O-sWL in the design event given by the bivariate approach in Figure 5 is 0.47m less than that in the existing approach for a 50-year return period, and the difference increases to 0.61m for the 100-year return period event. For return periods between 20 and 40 years both "surge only" and compound events arise, depending on the sample simulated from the fitted copulas.

Most-likely design events that are "surge only" (for smaller bivariate return periods) will potentially produce very different water levels at a structure (response variable) than compound events (for higher bivariate return periods), ultimately resulting in substantially different design conditions. For several flood defences in England, Gouldby et al. (2017) illustrated the sensitivity of the return period of a response variable – overtopping discharge – to the choice of return period definition. To account for the variability in design event selection, approaches have been developed to replace single design-events with


ensembles of possible design realizations (Gräler et al., 2013). Testing an ensemble of design events or adopting a structural based return period, where extremes are defined in terms of response variables directly, will produce a more robust analysis.

Implementation of these approaches would be particularly beneficial at sites S20 and S28, where, although all design events can be classified as "surge only" probability density is located along other parts of the isolines (see Figures SM.19 And SM.20). In many cases implementing these approaches requires running complex and computationally expensive process-based models, and is therefore beyond the scope of our analysis.



**Figure 5: Comparison of the design events (diamonds) obtained using the two-sided conditional sampling approach and the existing SFWMD approach (triangles) for return periods of (a) 10-, (b) 20-, (c) 50- and (d) 100-years. Quantile-isolines are superimposed onto plots of the observations, with blue circles (red crosses) denoting observations exceeding the rainfall (O-sWL) threshold and those exceeding neither threshold plotted in grey. Coloured contours signify the relative likelihood of events along an isoline, where the point with the highest density is selected as the most-likely**
**design event. Insets in (a) and (b) magnify the isoline about the associated most-likely design event.**

To further explore the conservative nature of the current design approach, the time for the O-sWL in the 100 design events for each return period derived using the bivariate analysis to reach the corresponding value under the full dependence assumption is quantified under different relative SLR scenarios. In other words, the amount of SLR and how long it will take under different emission scenarios, for the diamonds (i.e., bivariate design events) in Figure 5 to move vertically and close the gap to the

triangles (i.e., design events under full dependence assumption, currently used by SFWMD) is assessed. The *low*, *intermediate* and *high* scenarios from Sweet et al. (2017) are considered (see Figure 6, top).

**Figure 6: Top: Regional SLR projections for Miami Beach given in Sweet et al. (2017). Bottom: Number of years before the O-sWL in the 50-year design event derived using the bivariate approach reaches the corresponding value obtained**

**using the SFWMD approach according to the three SLR scenarios.**





The results are highly sensitive to the SLR scenario considered. For instance, the time before the O-sWL in the 50-year bivariate approach reaches that of the corresponding event derived using the SFWMD approach ranges from 16 years to greater than 80 years (Figure 6, bottom). The time before the O-sWL in the design events given by the SFWMD and bivariate approaches with return periods from 1 to 100 years become equal according to the three scenarios are shown in Figure 7. The

change in the characteristic of the design events (i.e., the shift from O-sWL dominated to compound driven) between return periods of around 20 to 40 years is apparent. For events with return periods > 40 years, the time before the O-sWL of the design events given by the two approaches coincide increases linearly with return period. According to the low SLR scenario the bivariate copula analysis combined with the "most-likely" design point suggests the currently employed assumption of full dependence between drivers is highly conservative, inadvertently incorporating safety factors sufficient to account for SLR

beyond the year 2100. Conversely, under the high SLR scenario the bivariate design assessment implies that the current approach is less conservative, with safety factors being exhausted within approximately 20 years for all return periods considered here (up to 100 years).

The disparity of the rainfall totals composing the design events given by the bivariate and existing approaches are greatest for low return periods (< 20 years), as demonstrated in Figure 5. It is possible that the rainfall totals will equate in the future due

to changes in rainfall patterns. However, rainfall projections were not examined here due their large uncertainties and lack of guidance regarding their coupling with SLR scenarios. Moreover, low return period events (< 20 years) are not typically used in structural design.

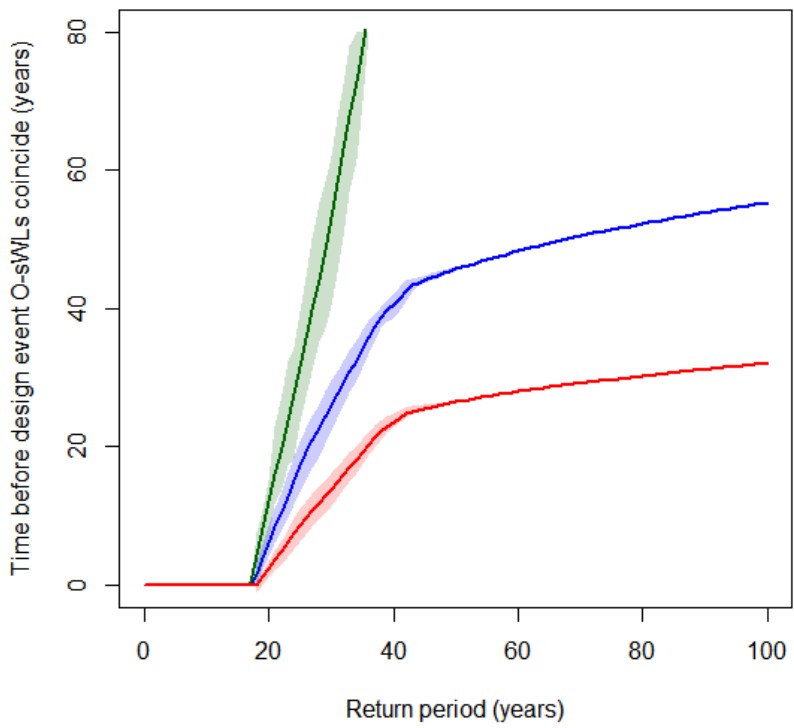





**Figure 7: Time before the O-sWL in the bivariate design event derived from the two-sided sampling approach reaches**
**corresponding value obtained from the existing approach (i.e., full dependence assumption) under the low (green),**
**intermediate (blue), and high (red) SLR scenarios given in Sweet et al. (2017). Shaded regions denote 95% (basic)**
**bootstrap confidence intervals.**

## 4.3 Trivariate analysis

In this section, the bivariate analysis is extended by also incorporating groundwater level into the analysis. First, the marginal
extremes are analysed separately for each flooding driver. The method of Smith and Weissman (1994) was applied to each
time series to identify cluster maxima. For each variable, cluster maxima and excesses above a sufficiently high threshold were
fitted to a GPD. The GPD was combined with the empirical distribution below the threshold. The threshold choice was guided
by appropriate criteria, predominantly mean residual life plots (Coles, 2001). Diagnostic goodness-of-fit demonstrated the
adequacy of the fit of the GPD for the rainfall and groundwater level series, whilst the fit to the O-sWL series was less robust
(see Figures SM.21 to SM.29). The study area is exposed to several flood-generating mechanisms including storms associated
with tropical cyclones, mesoscale convective systems, and extratropical systems. Hence, a single distribution is fitted to events
that are likely coming from several different populations. The fit of the GPD was particularly poor for the three largest O-sWL
events. The five-highest recorded O-sWL are associated with tropical cyclones, consistent with an analysis by Villarini and
Smith (2010). Nevertheless, observational records of the length available for this study contain relatively few tropical cyclone
events. Consequently, risk assessments in areas exposed to tropical cyclone storm surges commonly utilize synthetic records
of such events, generated based on historical observations (e.g., Nott, 2016). To generate synthetic records, wind and pressure
fields simulated from statistical models of tropical cyclone behaviours are used to drive hydrodynamic storm surge models
(Haigh et al., 2014). Replacing the observational record with a longer synthetic record could thus be an avenue to improve the
marginal fit of the O-sWL distribution, and ultimately the robustness of the proposed approach. This is beyond the scope of
the present study, where the focus is on developing appropriate frameworks for capturing and modelling dependence between
the different flood drivers.

The multivariate model fitting also requires sets of independent events. Gouldby et al. (2014) used a notional flooding level, a
function of the primary variables of interest, to de-cluster the offshore loading time series data before fitting the HT04 model.
In other applications of the HT04 approach, marginal de-clustered excesses of the conditioning variable are paired with
concurrent values of the remaining variables. The nonlinear regression model (Eq. 7) is then fitted to the set of events and the
process is repeated conditioning on each variable in turn. In the absence of a suitable response function that can be evaluated
without employing hydraulic/hydrologic models this is also the approach adopted here in the application of the HT04 model.
Standard higher dimensional copulas and vine copula models are often applied conditioning on a single variable to derive a
set of independent events. However, conditioning on only a single variable may result in the removal of the most extreme





values of the other variables. Therefore, in this work the models are applied to the entire dataset, as implemented before for higher dimensional copulas in Wong et al. (2008) and for vine copulas in Bevacqua et al. (2017), among others.

At all three sites, Gaussian and Student's *t* copula provided a similar fit in terms of AIC, far superior to that of the Archimedean copulas. The Gumbel copula was the only one of the considered Archimedean copulas to exhibit positive upper tail dependence

and resulted in the best fit among the three tested Archimedean copulas. For the three sites, scatter plots of the observations against 10,000 years worth of simulated data and Kendall's τ correlation coefficients indicate the vine copula offers a superior fit compared to the Gaussian copula, (see Figure 8 for the results at site S22 and Figures SM.30 and SM.31 for the corresponding plots at the other two sites). The plots also show that the HT04 model appears the most adept of the three approaches at capturing the dependence, particularly between O-sWL and the other variables.

The return periods conditional on a range of antecedent groundwater levels for the four bivariate (most-likely) design events, accounting for the dependence between rainfall and O-sWL (diamonds in Figure 5), according to the three types of trivariate models are shown in Figure 9. The trivariate return periods are calculated empirically from the samples in Figure 8. The bivariate events with return periods of 50- and 100- years were assigned return periods of >1000 years by the Gaussian and vine copulas for the groundwater levels considered and hence do not appear in Figure 9. In the case of the vine copula and

Gaussian copula, the 10- and 20- year bivariate design events exhibit sharp increases in return period about a narrow band of groundwater levels around 1 mNGVD 29. Given that the rainfall component of these bivariate design events is negligible (see upper row of Figure 5), the steep increases in return periods are consistent with the spike in simulations centred on this narrow band of groundwater levels seen in the two middle plots in the middle column of Figure 8. When extending the SFWMD design approach to include groundwater level, the annual exceedance event (i.e., trivariate event comprising the rainfall, Os-

WL, and groundwater level with univariate return periods of 1 year) possesses return periods of 2000, 227, and 116 years according to the Gaussian copula, vine copula, and HT04 approaches, respectively. Hence, differences between joint probabilities under the full dependence assumption and when accounting for actual dependencies increases further in the trivariate domain.





**Figure 8:** (1st row) **Observed events at site S22 (black dots) superimposed with the T-year return levels (grey lines) obtained from the marginal distributions and corresponding design events under the full dependence assumption (red dots). Kendall's τ coefficients are also displayed. Observed events (black dots) alongside 10,000-year synthetic event records (red dots) generated using the (2nd row) Gaussian copula, (3rd row) Vine copula, and (4th row) HT04 models.**

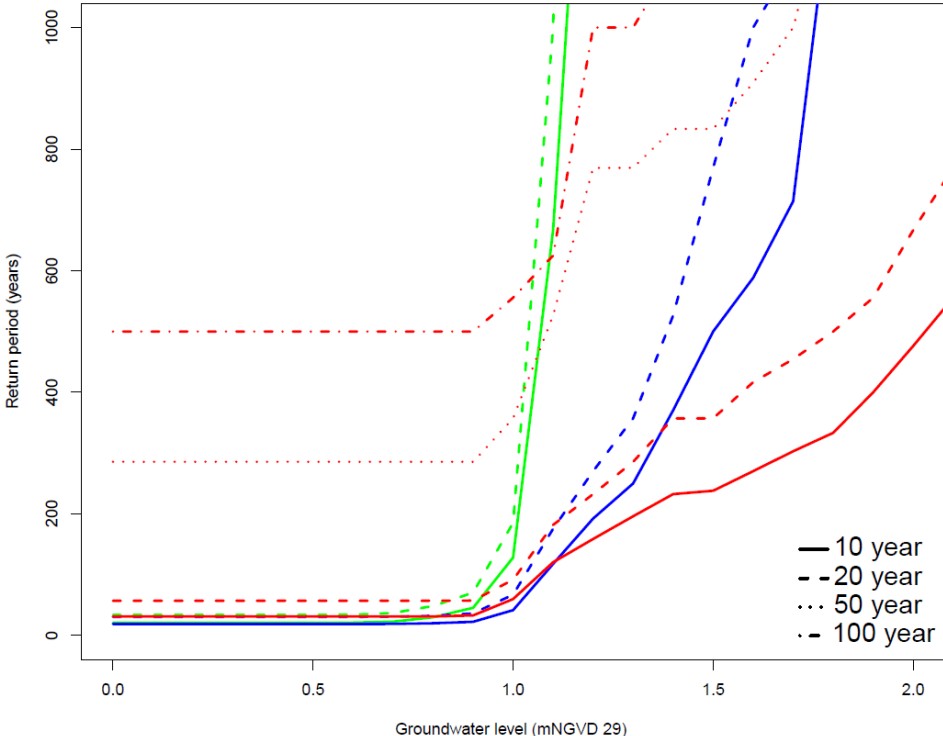


**Figure 9: Sensitivity of the return period of the four bivariate design events, derived using the approach described in Section 3.1 and displayed in Figure 4, to the antecedent catchment condition. The trivariate return periods are calculated using the Gaussian copula (green), vine copula (blue) and HT04 (red) approach.**

**5. Conclusions**

This paper puts forward a framework for assessing the different drivers of compound flooding in coastal areas of South Florida in Miami-Dade County. The framework was derived through a gradual transition from the current SFWMD structural design approach (based on the assumption of full dependence between rainfall and O-sWL and ignoring groundwater levels) by meeting three objectives. The first objective was to determine whether there is any statistically significant correlation between

extreme rainfall, O-sWL, and groundwater level in the area. At all three study sites rainfall, O-sWL, and groundwater level exhibit small but statistically significant pairwise correlations over a range of relevant time lags. The second objective was to assess the conservative nature of the present structural design approach that assumes full dependence. This was achieved by combining a bivariate analysis of the two flooding drivers with regional relative SLR scenarios. In the bivariate analysis, at site S22, low return period (< 20 year) design events constituted "surge only" events, hence the existing approach is deemed

highly conservative with respect to rainfall but less so in terms of O-sWL. The approach was shown to become ever more conservative in terms of O-sWL as return periods increase. The overall magnitude of the conservative assumption was found





to be highly dependent on the SLR scenario considered. For instance, any safety margin in the design according to the existing SFWMD approach for design events with return periods greater than 35 years is exhausted in less than 32 years under the high emission scenario. Conversely, for events with return periods up to 100 years this is expected to take more than 80 years under

the low emission scenario. At sites S20 and S28, although the bivariate design events for return periods 1- and 100-years were "surge events", non-negligible probability density was located along part of the isoline comprising compound events. The final objective was to provide robust estimates of the joint probabilities of extreme rainfall, O-sWL, and groundwater table for implementation in future structural design assessments. Three types of multivariate statistical models – five standard higher dimensional copulas, vine copula, and the HT04 model – were applied to capture the dependence structure in the extremes of

rainfall, O-sWL, and groundwater level. The vine copula and HT04 models better capture the dependence than any of the five-tested standard higher dimensional copulas.

The output of the bivariate and particularly trivariate applications can also act as boundary conditions for coupled hydrologic-hydraulic models for assessing flood risk and designing flood defence structures, among other purposes (e.g., Serafin et al., 2019). Rigorous implementation of the bivariate and trivariate methodologies, e.g., by adopting a structure-based return period

approach, or using an ensemble of events, will potentially facilitate more effective flood risk management in low-lying coastal catchments. A natural next step would be to explore the influence of the more robust boundary conditions on the design assessments of the flood defense assets at the three sites. Meanwhile, the accuracy of the GPD fit to O-sWL at the study sites (especially in the trivariate analysis, see Figures SM.22, SM.25 and SM.28) could also be improved by utilizing synthetic tropical cyclone events and associated storm surges. The methodologies introduced here are readily transferable and applicable

to other locations, assuming sufficiently long overlapping records of the different variables are available.

**Code availability**

Code and data used to complete this study are available in the *MultiHazard* R package which can be downloaded from GitHub at https://github.com/rjaneUCF/MultiHazard.

**Data availability**

See above. The data used in this paper are also freely available through NOAA's National Climatic Data Center's (NCDC) archive of global historical weather and climate data at https://www.ncdc.noaa.gov/cdo-web (rainfall) and SFWMD's corporate environmental database DBHYDRO at https://www.sfwmd.gov/science-data/dbhydro (O-sWL and groundwater level).



**Author contributions**

The study was conceived by JO and TW. RJ developed the methodology, undertook the analyses, and wrote the paper under the guidance of TW. LC and JO contributed by generating ideas, providing valuable insights during technical discussions and editing the manuscript.

**Competing interests**

TW is on the Editorial Board of this special issue.

**Acknowledgements**

RJ was supported by funding from the South Florida Water Management District. TW acknowledges financial support from the USACE Climate Preparedness and Resilience Community of Practice and Programs. This material is based in part on work supported by the National Science Foundation under Grant AGS-1929382. (T.W.).

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
