# Peer review of "Multivariate statistical modelling of the drivers of compound flood events in South Florida"

_Natural Hazards and Earth System Sciences, 2020_

## Referee Comment (RC1) · Anonymous Referee #1 · 16 May 2020

Authors have proposed a framework for assessing the compounding effects of flooding drivers in coastal areas of South Florida (e.g. rainfall, coastal water level (WL) and groundwater level). They first assess the significance of dependence between these variables, and then propose bivariate and trivariate approaches to generate compound design scenarios. They finally compare the generated design scenarios with the current design approaches (with the assumption of full dependence between variables). The idea is very interesting and manuscript is very well written. This work would be a significant and novel contribution to the community and deserves publishing in NHESS, however after a major revision. My main concerns are: i) continuity of probability density function along the hazard isolines must be checked, ii) generated bivariate and trivariate hazard scenarios are significantly inconsistent at the margin (rainfall-WL

plane) which needs a careful consideration. Details are provided below:

Major comments: - When the quantile-isoline is created by overlapping two separate isolines, could you please explain how you ensure that the joint PDF estimates are consistent along the envelope? I mean, theoretically taking derivative of CDF function, we are not necessarily dealing with a continuous function, and there is a chance that derivatives (that give us relative likelihood estimates and shading along the curve in Figure 5) diverge around the break point, right? Please, elaborate here. I see that you explained the process as "In this work, the relative probabilities are estimated non-parametrically via a Kernel Density Estimate (KDE), using the ks R package (Duong, 2007). Initially KDE was applied to the observations, however, particularly for larger return periods the design event proved highly sensitive to a small number of observations. Hence, design events were determined by applying KDE to a large sample ðİŚĄ = 10,000 from the two fitted copulas, with sample proportions consistent with the empirical distributions, and transformed back to original scales." But not yet clear to me how you check the continuity of PDF along the quantile-isoline. This probably helps better understand why "For small return periods ($\leq$20 years), design event rainfall remained <1mm, thus they may be considered "surge only" events." Looking at Figure 5a, we see some orange spectrum around the break point, while most likely scenario falls on the margin! You have also come up with the conclusion that "At sites S20 and S28, although the bivariate design events for return periods 1- and 100-years were "surge events", non-negligible probability density was located along part of the isoline comprising compound events" To me it raises a reddish flag that continuity is omitted.

- In page 18, where you analyze the timing before the O-sWL in the bivariate design event derived from the two-sided sampling approach reaches the corresponding value obtained from SFWMD scenarios (Figure 7), there is a significant underlying assumption that needs an explicit explanation, which is non-stationarity of the correlation structure. Indeed, the Copula parameters are assumed to remain unchanged over time.

- There is significant difference between compound scenarios using bivariate and

trivariate approaches. In Figures 5a and 5b, the design scenario is picked so close to the margin (with rainfall $\sim 0$) and a conclusion is made "In the bivariate analysis, at site S22, low return period (< 20 year) design events constituted "surge only" events". While in Figure 8 upper right panel, RP= 10 yr comes with $\sim$200 mm of rainfall. Also, in Figure 5c the pair associated with RP=50 yr is (Rainfall = 185, WL = 1.11), while in Figure 8 (upper right panel) the pair associated with RP = 50 yr is (Rainfall = 340, WL = 1.778) that is significantly larger than the one proposed under bivariate analysis. I understand that sampling approach is different between cases, but distribution functions must be compatible at the margins (i.e. at Rainfall-WL plane) otherwise yields in a great confusion. Especially when you state "The output of the bivariate and particularly trivariate applications can also act as boundary conditions for coupled hydrologic/hydraulic models for assessing flood risk and designing flood defence structures". Also, bivariate approach is based on extreme two-sided sampling, which is a more realistic approach compared with the one under trivariate analysis. So, does this significant different between outcomes suggest inappropriateness of employed trivariate scheme with the aim of "accounting for actual dependencies"?

Minor comments: L180-185: Sampling specifications not provided. L223: Define the variables used in this equation. L496: have not defined "mNGVD"

Nice work and good luck!

---

## Referee Comment (RC2) · Anonymous Referee #2 · 18 May 2020

The authors apply multivariate statistical analysis approaches to assess the correlation between flood drivers, particularly rainfall, ocean-side water levels, and groundwater levels, in South Florida. They then evaluate existing structural design approaches considering compound rainfall and surge and the effects of sea-level rise. Finally, they apply higher-dimensional copulas to generate estimates of joint probabilities between the three flood drivers. Overall, the paper is well-researched and written and applies a robust statistical analysis approach. It advances past assessments of compound flood drivers and is relevant to the scope of NHESS. Prior to acceptance, I recommend further assessing the groundwater contribution to compound events and strengthening the discussion of how the results of this analysis can inform planning/management.

Specific comments:

[Figure]

-When groundwater is incorporated, you find that "the annual exceedance event (i.e., trivariate event comprising the rainfall, Os-WL, and groundwater level with univariate return periods of 1 year) possesses return periods of 2000, 227, and 116 years" (L499). While it is important to note the likelihood of co-occurrence of these three exceedance events, co-occurrence of a high groundwater table and heavy rainfall OR extreme O-sWL is also a concern for flood management. The results of bivariate analysis of these interactions would provide further insight into the potential mechanisms of flooding in the region.

-You mention that rainfall cluster maxima "are paired with simultaneous O-sWL values and vice versa" (L185). Did you consider different time lags across the three sites?

-It would be helpful to have more information about SFWMD's planning/design approach and how groundwater levels are considered. What types of structures are designed using the full-dependence approach? Does SFWMD have existing thresholds for groundwater levels that are used in the design or operation of their facilities? Are there seasonal differences in how the system is managed given rainfall patterns and the need to limit salt-water intrusion?

-You state in the abstract that this analysis "leads to recommendations for revised future design frameworks able to capture and represent dependencies between different flood drivers," but you provide little discussion of how this information could be incorporated into SFWMD's planning or what changes would be appropriate given the study results. How should the design guidelines be modified, if at all, especially considering future sea-level rise?

Technical corrections:

-The abstract should include more information about the results obtained.

-L26: No need to capitalize "state".

-L35: Miami is spelled incorrectly.

[Figure]

-L411: Rephrase "probability density is located along other parts of the line". For example, you could say "probability density is non-zero [or above a certain threshold] along other parts of the line."

-L421: This sentence is a bit confusing.

-L441: Looks closer to 30 years for the 100-year return period.

---

## Author Comment (AC1) · 10 Jul 2020

**Response to Anonymous Referee 1**

**Authors have proposed a framework for assessing the compounding effects of flooding drivers in coastal areas of South Florida (e.g. rainfall, coastal water level (WL) and groundwater level). They first assess the significance of dependence between these variables, and then propose bivariate and trivariate approaches to generate compound design scenarios. They finally compare the generated design scenarios with the current design approaches (with the assumption of full dependence between variables). The idea is very interesting and manuscript is very well written. This work would be a significant and novel**

**contribution to the community and deserves publishing in NHESS, however after a major revision. My main concerns are: i) continuity of probability density function along the hazard isolines must be checked, ii) generated bivariate and trivariate hazard scenarios are significantly inconsistent at the margin (rainfall-WL plane) which needs a careful consideration. Details are provided below:**

The authors would like to thank the reviewer for providing thoughtful comments. Please find our responses to each comment below including how we plan (following the NHESS review process) to adjust the manuscript.

**Major comments:**

**- When the quantile-isoline is created by overlapping two separate isolines, could you please explain how you ensure that the joint PDF estimates are consistent along the envelope? I mean, theoretically taking derivative of CDF function, we are not necessarily dealing with a continuous function, and there is a chance that derivatives (that give us relative likelihood estimates and shading along the curve in Figure 5) diverge around the break point, right? Please, elaborate here. I see that you explained the process as "In this work, the relative probabilities are estimated nonparametrically via a Kernel Density Estimate (KDE), using the ks R package (Duong, 2007). Initially KDE was applied to the observations, however, particularly for larger return periods the design event proved highly sensitive to a small number of observations. Hence, design events were determined by applying KDE to a large sample = 10,000 from the two fitted copulas, with sample proportions consistent with the empirical distributions, and transformed back to original scales." But not yet clear to me how you check the continuity of PDF along the quantile-isoline. This probably helps better understand why "For small return periods (20 years), design event rainfall remained <1mm, thus they may be considered "surge only" events." Looking at Figure 5a, we see some orange spectrum around the break point, while most likely scenario falls on the mar-**

**gin! You have also come up with the conclusion that "At sites S20 and S28, although the bivariate design events for return periods 1- and 100-years were "surge events", non-negligible probability density was located along part of the isoline comprising compound events" To me it raises a reddish flag that continuity is omitted.**

The derivation of the isolines and the estimation of the contours are carried out independently. For the latter, as discussed in the manuscript and quoted above, a kernel density estimate is used to estimate the joint PDF of the observations (approximated by large samples from the two fitted copulas). Once a given isoline is derived the probability density associated with pairs of rainfall and O-sWL falling on the isoline are extracted (and scaled to lie within $[0, 1]$, hence relative probabilities). Therefore, no differentiation of the CDF is required and continuity is preserved. To clarify that no differentiation of the CDF is required as the probabilities are obtained directly from the KDE the following sentence has been added to L387:
*"The probability density given by the KDE at points along the isoline are extracted and the probabilities scaled to lie within $[0, 1]$, hence yielding relative probabilities."*

**- In page 18, where you analyze the timing before the O-sWL in the bivariate design event derived from the two-sided sampling approach reaches the corresponding value obtained from SFWMD scenarios (Figure 7), there is a significant underlying assumption that needs an explicit explanation, which is non-stationarity of the correlation structure. Indeed, the Copula parameters are assumed to remain unchanged over time.**

The authors agree this is an important assumption which warrants an explicit explanation in the text. The following passage has been added to the manuscript.
*"Nonstationarity in the dependence between rainfall and O-sWL can occur as a consequence of a range of anthropogenic and climatic induced stressors. In this study, the dependence is assumed stationary i.e., that the copula parameters remain unchanged over time. The overlapping records at the three sites are of insufficient length to ro-*

*bustly test the stationarity assumption. However, Wahl et al. (2015) did not detect any significant change in Kendall's τ between rainfall and surge at either Key West or Mayport, the two closest sites to Miami-Dade County, indicating stationarity may be a reasonable assumption. Nevertheless, due to regional and local effects, such as multi-decadal variation in the storm surge climate, the possibility of statistically significant trends in the dependence cannot be ruled out at the case study sites."*

**There is significant difference between compound scenarios using bivariate and trivariate approaches. In Figures 5a and 5b, the design scenario is picked so close to the margin (with rainfall approx. 0) and a conclusion is made "In the bivariate analysis, at site S22, low return period ($<$ 20 year) design events constituted "surge only" events". While in Figure 8 upper right panel, RP= 10 yr comes with 200 mm of rainfall. Also, in Figure 5c the pair associated with RP=50 yr is (Rainfall = 185, WL = 1.11), while in Figure 8 (upper right panel) the pair associated with RP = 50 yr is (Rainfall = 340, WL = 1.778) that is significantly larger than the one proposed under bivariate analysis. I understand that sampling approach is different between cases, but distribution functions must be compatible at the margins (i.e. at Rainfall-WL plane) otherwise yields in a great confusion. Especially when you state "The output of the bivariate and particularly trivariate applications can also act as boundary conditions for coupled hydrologic/hydraulic models for assessing flood risk and designing flood defence structures". Also, bivariate approach is based on extreme two-sided sampling, which is a more realistic approach compared with the one under trivariate analysis. So, does this significant different between outcomes suggest inappropriateness of employed trivariate scheme with the aim of "accounting for actual dependencies"?**

Thank you for this very useful comment. As you correctly state the sampling approaches differ between the bivariate and trivariate analyses. The purpose of the bivariate analysis is to demonstrate the conservative nature of the full dependence assumption between rainfall and O-sWL used in the original design of the structures.

To ensure only the impact of accounting for the dependence is being assessed, the two design events (triangles and diamonds) are derived using identical marginals. The fact that the marginal distributions are the same is evidenced in Figure 5 by the aligning of the design events derived under the assumption of full dependence (triangles) with the isolines at the margins. However, the disadvantage of using identical marginals is that the design events derived under the assumption of full dependence in Figure 5 may not be equivalent to those derived using the original design approach as that approach did not use the two-sided conditional sampling. To make this clear reference to *"the current design approach"* in this section have been changed to *"the assumption of full dependence"*.

In the trivarite analysis section the design events under the assumption of full dependence are shown in the top row of Figure 8, thus the design events in the upper right panel are likely to be more reflective of those adopted by the SFWMD using their current level of service assessments. In relation to the comparisons made in the comment, the events in the upper right panel of Figure 8 would be more comparable to the diamonds in Figure 5 rather than the triangles, however, they were derived using different marginal distributions and so any comparison must be made with care. The design events derived under the assumption of full dependence (first row of Figure 8) and realizations from the trivariate models (remaining rows of Figure 8) are obtained using identical marginal distributions, hence the distribution functions are compatible at the margins and the effect of accounting for the dependence can be assessed. Nevertheless, this was not explicitly discussed in the original manuscript, therefore the following sentence has been added on L489 to address this.

*"Overall the sparsity of simulation data near the design events (with return periods greater than 1-year) obtained under the assumption of full dependence demonstrates the importance of accounting for the dependencies between the drivers when assessing the compound flood hazard."*

A comparison between the bivariate and trivariate results using the new methods

(conditional sampling in the two-dimensional case and copulas/HT04 in the three-dimensional case) would involve comparing the bivariate design events i.e., diamonds in Figure 5, with the realizations of rainfall-O-sWL from the trivariate models i.e., red dots in the 2nd-4th row of the 3rd column of Figure 8 (as opposed to the first row of Figure 8). As correctly stated in the comment the bivariate approach yields "surge only" events for lower return periods at site S-22. Realizations from the trivariate models with low O-sWL (say 1mNGVD) and zero rainfall are common particularly for the Gaussian and vine copulas in agreement with the bivariate results. For the HT04 method which is arguably the most flexible and thus robust of the trivariate methods, as the O-sWL increases events with zero rainfall become increasingly rare. The HT04 model thus implies higher return period events are more likely to become compound in nature as return period increases, again, in common with the bivariate results. The latter trend is less pronounced for the Gaussian and vine copulas nevertheless the disparity between the bivariate and trivariate results is perhaps not as great as implied in the comment. As correctly stated in the comment sampling approaches are different between the (bivarite and trivariate) cases and the distribution functions must be compatible at the margins if the ability of the bivariate and trivarite approaches to capture the dependence are to be compared. Furthermore, the two cases are addressing different issues consequently we refrain from comparing the two cases in the manuscript.

To summarize the response so far, both the trivariate and bivariate cases the marginal distributions used to derive the design events with/without dependence are identical in order isolate the effect of modeling the dependence.

**Minor comments:**

**L180-185: Sampling specifications not provided.**
Thank you for the comment. The first sentence of Section 3.2 has been expanded to give more details about the sampling specifications and now reads as follows:

*"Here, a two-sided sampling approach similar to that in Wahl et al. (2015), which*

[Figure]

*involves deriving two conditional samples where each variable is conditioned on in turn, is implemented to identify bivariate extreme events."*

In addition, a reference to the results section where more details on the choice of thresholds can be found has also been added to the end of the paragraph starting on L182:

*"For more details on the choice of thresholds see Section 4.2."*

**L223: Define the variables used in this equation.**
The variables are now defined in the text.

**L496: have not defined "mNGVD"**
Thank you for pointing this out, a definition of mNGVD has been added.

---

## Author Comment (AC2) · 10 Jul 2020

**Response to Anonymous Referee 2**

he authors apply multivariate statistical analysis approaches to assess the correlation between flood drivers, particularly rainfall, ocean-side water levels, and groundwater levels, in South Florida. They then evaluate existing structural design approaches considering compound rainfall and surge and the effects of sea-level rise. Finally, they apply higher-dimensional copulas to generate estimates of joint probabilities between the three flood drivers. Overall, the paper is well-researched and written and applies a robust statistical analysis approach. It advances past assessments of compound flood drivers and is relevant

[Figure]

**to the scope of NHESS. Prior to acceptance, I recommend further assessing the groundwater contribution to compound events and strengthening the discussion of how the results of this analysis can inform planning/management.**

Thank you for the pertinent comments. Please find our responses to each comment below including how we plan (following the NHESS review process) to adjust the manuscript.

**Specific comments:**

**-When groundwater is incorporated, you find that "the annual exceedance event (i.e., trivariate event comprising the rainfall, Os-WL, and groundwater level with univariate return periods of 1 year) possesses return periods of 2000, 227, and 116 years" (L499). While it is important to note the likelihood of co-occurrence of these three exceedance events, co-occurrence of a high groundwater table and heavy rainfall OR extreme OsWL is also a concern for flood management. The results of bivariate analysis of these interactions would provide further insight into the potential mechanisms of flooding in the region.**

Agreed, plots of the high groundwater table and heavy rainfall OR extreme OsWL associated with return periods of 10-,20-,50- and 100-years at site S22 obtained using the 10,000-year synthetic event records from the three approaches have been added to the supplementary material (see Figures 1 and 2 in this document, dotted lines represent extrapolation). The following comment has been added to the text to summarize the results:

*"Similar patterns emerge when considering the co-occurrence of groundwater level and either rainfall or O-sWL, see Figures SM.32 and SM.33."*

**-You mention that rainfall cluster maxima "are paired with simultaneous O-sWL values and vice versa" (L185). Did you consider different time lags across the three sites? No time lags were considered at any of the sites in the multivariate**

**statistical modeling.**

Thank you for highlighting that this was not stated as clearly as it should have been in the original manuscript. Please see the following paragraph which has been added to Section 3.1 detailing our justification for why no lag was considered in the statistical modelling.

*"To ensure that temporally coherent combinations of the drivers are simulated no lags are considered in either the bivariate or trivariate analysis. For instance, applying a lag to the groundwater level will account for its maximum correlation with O-sWL and rainfall at sites S22 and S28. However, by the time the elevated groundwater level arises the high O-sWL may have dissipated and rainfall potentially ceased, thus it is possible the drivers do not produce any compounding effects."*

**-It would be helpful to have more information about SFWMD's planning/design approach and how groundwater levels are considered. What types of structures are designed using the full-dependence approach? Does SFWMD have existing thresholds for groundwater levels that are used in the design or operation of their facilities? Are there seasonal differences in how the system is managed given rainfall patterns and the need to limit salt-water intrusion?**

Thank you for this thoughtful comment. Please see the following two paragraphs which will be added to the introduction to more thoroughly explain the original design and current work being undertaken by SFWMD to assess the level of service provided by the relevant structures.

*"Water control facilities for the Central and South Florida Project (CSFP) authorized by the Flood Control Act of June 30, 1948 (Pub. L. 80-858, 46 Stat. 925) were designed by the USACE in the 1950s and 60s. The project included hydrologic and hydraulic design for canals, many of which terminate in flood/salinity control structures. The control structures are operated by the SFWMD to maintain the water level to prevent saltwater intrusion and release canal water to the sea (typically via tidally modulated channels)*

*alleviating potential flooding. The design of the canal required pairing a design O-sWL, typically obtained from tide tables, and a design storm under the assumption of full dependence; i.e. bivariate design event associated with a return period is obtained by pairing the O-sWL and peak rainfall with the corresponding univariate return periods. Groundwater level conditions were accounted for through the rainfall input. For instance, in the Greater Miami area, it was assumed that the first four inches of rainfall of the design storm would be used to replenish the groundwater storage.*

*The SFWMD is beginning to revisit the original designs of coastal water control structures. The SFWMD's Flood Protection Level of Service (FPLOS) project is examining the protection that existing coastal structures provide to urban areas, adopting a more holistic approach as compared to 1950s and 60s designs. FPLOS uses design storms, which are run through hydrologic models with initial conditions given by groundwater stages. For coastal structures, the O-sWL represents an additional downstream boundary condition described by a stage hydrograph. Peak stages in the boundary condition hydrographs are derived using frequency analysis, hence in FPLOS assessments rainfall, O-SWL and groundwater level are assumed fully dependent."*

Reference: Flood Control Act of 1948, Pub. L. 80-858, 46 Stat.925, 1948.

Other passages such as: *"structures, operated by SFWMD to maintain the water level to prevent saltwater intrusion and release canal water to the sea (typically via tidally modulated channels) alleviating potential flooding."* (L118-119) are removed to avoid repetition.

With regards to seasonality, the design of stormwater control facilities submitted as part of the Environmental Permitting requirements at the SFWMD are required to consider high seasonal groundwater stages in the hydrologic and hydraulic design. These more typically occur during the wet season period of June through October.

**-You state in the abstract that this analysis "leads to recommendations for revised future design frameworks able to capture and represent dependencies be-**

**tween different flood drivers," but you provide little discussion of how this information could be incorporated into SFWMD's planning or what changes would be appropriate given the study results. How should the design guidelines be modified, if at all, especially considering future sea-level rise?**

The authors agree that the phrase used in the abstract is not consistent with the content of the remainder of the manuscript. The work represents the first steps towards the development of a new framework that accounts for the dependencies between flooding drivers as part of assessing the level of service provided by coastal water control structures rather than providing explicit recommendations. To reflect these points the sentence quoted in the comment above has been rephrased as follows:

*"The work represents the first steps towards the development of a new framework capable of capturing dependencies between different flood drivers that could potentially be incorporated into future FPLOS assessments for coastal water control structures."*

The authors are happy to report a second phase of this project has recently been funded by SFWMD. The second phase will see District staff given tutorials to further their understanding of the statistical techniques used in the new framework and additional training on how to use the R package, thus representing the next step in potentially modifying design/risk analysis guidelines.

**Technical corrections:**

**-The abstract should include more information about the results obtained.**
The abstract has been amended to provide more details on the results obtained. For instance, the sentence starting on L16 now reads:
*"A two-dimensional analysis of rainfall and O-sWL showed that the magnitude of the conservative assumption in the existing structural design assessment is highly sensitive to the regional sea-level rise projection considered."*

**-L26: No need to capitalize "state".**

Thank you for the comment. The grammar been corrected.

**-L35: Miami is spelled incorrectly.**
Thank you. The spelling has been corrected.

**-L411: Rephrase "probability density is located along other parts of the line".
For example, you could say "probability density is non-zero [or above a certain
threshold] along other parts of the line."**
Agreed, the text has been changed in line with the recommendation.

**-L421: This sentence is a bit confusing.**
The sentence has been amended and now reads as follows: *"The conservative nature
of the current design approach is further explored by assessing how long it will take un-
der a given SLR for the 100-year design events selected with the two different methods
(i.e. full dependence assumption vs bivariate dependence modelling) to intersect."*

**-L441: Looks closer to 30 years for the 100-year return period.**
Correct, the text has therefore been amended to reflect this. Thank you for pointing this
out.
* * *
2020-82, 2020.

**Fig. 1.** Isolines of rainfall and groundwater level from the Gaussian copula (blue), Vine copula (green) and HT04 model (red) for return periods (a) 10- (b) 20- (c) 50- and (d) 100- years.

**Fig. 2.** Isolines of O-sWL and groundwater level from the Gaussian copula (blue), Vine copula (green) and HT04 model (red) for return periods of (a) 10- (b) 20- (c) 50- and (d) 100- years.